# Alcohol Metabolism Enriches Squamous Cell Carcinoma Cancer Stem Cells That Survive Oxidative Stress via Autophagy

**DOI:** 10.3390/biom11101479

**Published:** 2021-10-07

**Authors:** Masataka Shimonosono, Koji Tanaka, Samuel Flashner, Satoshi Takada, Norihiro Matsuura, Yasuto Tomita, Uma M. Sachdeva, Eishi Noguchi, Veena Sangwan, Lorenzo Ferri, Fatemeh Momen-Heravi, Angela J. Yoon, Andres J. Klein-Szanto, J. Alan Diehl, Hiroshi Nakagawa

**Affiliations:** 1Herbert Irving Comprehensive Cancer Research Center, Columbia University Irving Medical Center, New York, NY 10032, USA; k5541938@kadai.jp (M.S.); sf3070@cumc.columbia.edu (S.F.); s.takada.med@gmail.com (S.T.); nm3281@cumc.columbia.edu (N.M.); yt2795@cumc.columbia.edu (Y.T.); Uma.Sachdeva@mgh.harvard.edu (U.M.S.); fm2540@cumc.columbia.edu (F.M.-H.); ajk55@cumc.columbia.edu (A.J.Y.); 2Department of Gastroenterological Surgery, Graduate School of Medicine, Osaka University, Osaka 565-0871, Japan; ktanaka@gesurg.med.osaka-u.ac.jp; 3Department of Surgery, Division of Thoracic Surgery, Massachusetts General Hospital, Boston, MA 02114, USA; 4Department of Biochemistry and Molecular Biology, Drexel University College of Medicine, Philadelphia, PA 19102, USA; en34@drexel.edu; 5Department of Surgery, Montreal General Hospital, McGill University, Montreal, QC H3G 1A4, Canada; veena.sangwan@gmail.com (V.S.); lorenzo.ferri@mcgill.ca (L.F.); 6Cancer Biology and Immunology Laboratory, College of Dental Medicine, Columbia University Irving Medical Center, New York, NY 10032, USA; 7Department of Pathology & Cell Biology, Division of Oral & Maxillofacial Pathology, Columbia University Irving Medical Center, New York, NY 10032, USA; 8Histopathology Facility, Fox Chase Cancer Center, Philadelphia, PA 19111, USA; Andres.Klein-Szanto@fccc.edu; 9Case Comprehensive Cancer Center, Department of Biochemistry, School of Medicine, Case Western Reserve University, Cleveland, OH 44106, USA; jad283@case.edu; 10Department of Medicine, Division of Digestive and Liver Diseases, Columbia University Irving Medical Center, New York, NY 10032, USA

**Keywords:** alcohol, autophagy, CD44, organoids, squamous cell carcinoma

## Abstract

Background: Alcohol (ethanol) consumption is a major risk factor for head and neck and esophageal squamous cell carcinomas (SCCs). However, how ethanol (EtOH) affects SCC homeostasis is incompletely understood. Methods: We utilized three-dimensional (3D) organoids and xenograft tumor transplantation models to investigate how EtOH exposure influences intratumoral SCC cell populations including putative cancer stem cells defined by high CD44 expression (CD44H cells). Results: Using 3D organoids generated from SCC cell lines, patient-derived xenograft tumors, and patient biopsies, we found that EtOH is metabolized via alcohol dehydrogenases to induce oxidative stress associated with mitochondrial superoxide generation and mitochondrial depolarization, resulting in apoptosis of the majority of SCC cells within organoids. However, CD44H cells underwent autophagy to negate EtOH-induced mitochondrial dysfunction and apoptosis and were subsequently enriched in organoids and xenograft tumors when exposed to EtOH. Importantly, inhibition of autophagy increased EtOH-mediated apoptosis and reduced CD44H cell enrichment, xenograft tumor growth, and organoid formation rate. Conclusions: This study provides mechanistic insights into how EtOH may influence SCC cells and establishes autophagy as a potential therapeutic target for the treatment of EtOH-associated SCC.

## 1. Introduction

Chronic alcohol consumption poses increased risks for many cancer types [1]. The foremost organ sites linked to a strong alcohol-related cancer risk are the mouth, tongue, throat and the esophagus [2,3] where squamous cell carcinoma (SCC) represents the major tumor type. SCC of the head and neck (HNSCC) and the esophagus (ESCC) are common worldwide, and are deadly due to late diagnosis, metastasis, therapy resistance, and early recurrence [4,5]. HNSCC and ESCC develop on the mucosal surface that is directly exposed to high concentrations of ethanol (EtOH) during alcohol consumption [6,7]. Additionally, SCC cells may be influenced by EtOH and EtOH metabolites in circulation.

EtOH is a major human carcinogen; however, how EtOH promotes tumorigenesis is incompletely understood [8]. EtOH exerts genotoxic effects through induction of DNA adducts, DNA damage, and oxidative stress, resulting in increased epithelial cell proliferation in oral and esophageal mucosa [9]. In normal human esophageal epithelial cell lines, cytochrome P450 2E1 and alcohol dehydrogenase (ADH) 1B catalyze EtOH oxidation, which generates acetaldehyde, a toxic metabolite that induces cell injury by perturbing mitochondrial respiration and the electron transportation chain, causing oxidative stress and apoptosis [10]. However, how SCC tumor cells respond to EtOH exposure remains elusive.

HNSCC and ESCC are characterized by intratumoral cell heterogeneity [11,12]. Amongst cancer cells are a unique subset referred to as cancer stem cells (CSCs) or tumor-initiating cells with high expression of cell-surface CD44 (CD44H) glycoprotein. CD44H cells display increased malignant properties including invasion, metastasis, and therapy resistance in addition to a high tumor-initiation capability [13,14,15,16,17,18,19,20]. Although alcohol has been shown to induce CSCs in breast and liver cancers [21,22], how SCC cells react to EtOH exposure has not been studied.

We have recently developed a novel three-dimensional (3D) oral and esophageal organoid system where single cell-derived normal and neoplastic epithelial structures recapitulate the morphology, gene expression, and functions of the original tissue [23,24]. 3D organoids generated from SCC patients and cell lines contain CD44H cells where reproduced chemotherapy resistance is in part mediated by autophagy [23], the evolutionarily conserved cytoprotective mechanism that degrades and recycles damaged and dysfunctional cellular organelles such as mitochondria.

In this study, we have evaluated the effect of EtOH exposure in SCC 3D organoids and xenograft tumors. We found that EtOH metabolism in SCC cells leads to oxidative stress, mitochondrial dysfunction, and apoptosis of non-CD44H cells, permitting enrichment of CD44H cells that survive via autophagy.

## 2. Materials and Methods

### 2.1. Cell Culture and 3D Organoid Culture

All cell culture equipment and reagents were purchased from Thermo Fisher Scientific (Waltham, MA, USA) unless otherwise noted. The number of live cells in culture or tissues were determined by Countess^™^ Automated Cell Counter coupled with 0.2% Trypan Blue dye staining test to exclude dead cells. ESCC cell lines TE11 (a gift of Dr. Tetsuro NIshihira, Tohoku University School of Medicine, Sendai, Miyagi, Japan) and TE14 (RCB2101; Cellosaurus Expasy CVCL_3336) (RIKEN BioResource Research Center Cell Engineering Division/Cell Bank, Tsukuba, Ibaraki, Japan) [25] and genetically modified derivatives were grown in monolayer culture in RPMI-1640 supplemented with 10% fetal bovine serum and penicillin (100 units/mL)-streptomycin (100 µg/mL) and utilized to generated three-dimensional (3D) organoids as described previously [23,24].

Two independent ESCC patient-derived organoid (PDO) lines, ESC2 and ESC3, were established from endoscopic ESCC tumor biopsies [23,24] that were obtained via upper endoscopy at the McGill University (VS and LF). Cryopreserved HNSCC patient-derived xenograft (PDX) tumors OCTT2, OCTT79, and HPPT7 [26] were utilized to establish HNSCC PDO lines HSC1, HSC2, and HSC3, respectively. All clinical materials were procured from informed-consent patients according to the Institutional Review Board standard and guidelines.

To generate 3D organoids, a single cell suspension was seeded into 24-well plates with 500–2000 cells mixed in 50 µL Corning™ Matrigel^®^ Basement Membrane Matrix (#354234) per well, and grown for 11 days with 500 µL (per well) of advanced DMEM/F12 medium supplemented with 1X GlutaMAX™ Supplement, 10 mM HEPES, 1X N2 Supplement, 1X B27 Supplement, 1 mM *N*-acetyl-l-cysteine (NAC) (Sigma-Aldrich, St Louis, MO, USA), 50 ng/mL human recombinant epidermal growth factor (Peprotech, Cranbury, NJ, USA), 2.0% Noggin/R-Spondin-conditioned media [24] and 10 μM Y27632 (Selleck Chemicals, Houston, TX, USA). To monitor organoid growth in each well, phase-contrast images were captured by EVOS FL Cell Imaging System and bright-field images were captured by KEYENCE Fluorescence Microscope BZ-X800 (Keyence, Osaka, Osaka, Japan) equipped with the Keyence software, the latter utilized to determine number and size of individual organoids (defined as ≥5000 µm^2^ spherical structures). When indicated, cell viability for growing 3D organoids was evaluated by CellTiter-Glo^®^ 3D Cell viability assays (Promega, Fitchburg, WI, USA) according to manufacturer’s instructions.

### 2.2. EtOH Treatment, Pharmacological and Genetic Modifications of Cells

Cells were treated with 0.1–2% (*v*/*v*) EtOH diluted in growth medium as described [10]. To maintain alcohol saturation, the wells of cell culture plates were sealed with PARAFILM^®^ M (Sigma-Aldrich). Cells were concurrently treated with or without 2 mM 4-methylpyrazole (4MP) hydrochloride (Sigma-Aldrich), the ADH inhibitor, reconstituted in phosphate-buffered saline (PBS) at 100 mM. Alternatively, cells were concurrently treated with or without 2 µM chloroquine diphosphate (CQ) (Sigma-Aldrich), the autophagy flux inhibitor, as described previously [10,15].

Lentivirus-mediated gene transduction and puromycin selection of ESCC cell lines TE11 and TE14 was performed as described previously [13,19,27]. Cells were infected with lentivirus carrying doxycycline (DOX)-inducible short hairpin RNA (shRNA) directed against *ATG7* (pTRIPZ-ATG7, V2THS_262187 and V2THS_13452, two independent shRNA sequences) or a nonsilencing scrambled sequence (pTRIPZ-NS, RHS4346) (Dharmacon, Lafayette, CO, USA). shRNA was induced in culture with 1 μg/mL DOX as described [13,19,27]. TE11 and TE14 derivatives expressing constitutive turboRFP fluorescent protein (TE11-RFP and TE14-RFP) were established as described previously [13] to distinguish ESCC cells from host-derived stromal cells in xenograft transplantation experiments.

### 2.3. Flow Cytometry to Evaluate Cell Surface CD44 Expression, Proliferation, Apoptosis, Mitochondrial Mass, and Mitochondrial Membrane Potential

Flow cytometry and fluorescence-activated cell sorting (FACS) were performed as described previously [10,15,16,19] using LSR Fortessa or Influx cell sorter (BD Biosciences, Franklin Lakes, NJ, USA). Organoids were disintegrated and dissociated into single-cell suspensions as described [24]. Cells were washed once with PBS, resuspended in PBS containing 1% bovine serum albumin (Sigma-Aldrich), and stained with either allophycocyanin-conjugated anti-CD44 antibody (clone G44-26) (1:20; 559942; BD Biosciences) or phycoerythrin-conjugated anti-CD44 antibody (clone G44-26) (1:10; 555479; BD Biosciences) at 4 °C for 30 min to determine CD44 expression. Dead cells were detected by 4′,6-diamidino-2-phenylindole (DAPI) (Thermo Fisher Scientific) except for apoptosis assays where propidium iodide (PI) (640914, Biolegend, San Diego, CA, USA) was utilized. Flow cytometry data was analyzed using FlowJo software v10.7.1 (Tree Star, Ashland, OR, USA). Cells with high CD44 expression (CD44H) and low expression (CD44L) were identified as the top 10% and the bottom 10% of CD44 expressing cells, respectively. The CD44 gate was set in control organoids grown for 11 days without EtOH or drug treatment, and applied to all experimental groups to determine CD44L and CD44H cells in 3D organoids that were treated with EtOH concurrent with or without 4MP, CQ or DOX.

Proliferation of cells within 3D organoids was evaluated using Click-iT Plus EdU Alexa Fluor 488 Flow Cytometry Assay Kit (C10632, Thermo Fisher Scientific) according to manufacturer’s instructions. In brief, organoids were incubated with 10 µM Click-iT Plus EdU reagent for 2 h in culture and dissociated into a single-cell suspension. Cells were fixed, permeabilized, and incubated with Alexa Fluor 488 picolyl azide for 30 min at room temperature followed by flow cytometry to detect EdU-incorporated cells.

Apoptosis in CD44L and CD44H cells was determined by FITC Annexin V Apoptosis Detection Kit (Thermo Fisher Scientific) according to the manufacturer’s instructions as described previously [10,28].

Mitochondrial mass and membrane potential in CD44L and CD44H cells were measured with MitoTracker™ Green (MTG) (1:20,000; M7514; Thermo Fisher Scientific) and MitoTracker™ Deep Red (MTDR) (1:50,000; M22426; Thermo Fisher Scientific), respectively, as described previously [10,29]. 

Autophagy in CD44L and CD44H cells was determined by autophagosomes (AV)-identifying Cyto-ID^®^ fluorescent dye (ENZ-51031; Enzo Life Sciences, Farmingdale, NY, USA) as previously described [10,15,28,29]. 

### 2.4. Mitochondrial Superoxide Assays

To determine mitochondrial superoxide, cells in monolayer culture were incubated with 5 µM of MitoSOX™ (M36008; Thermo Fisher Scientific) and 20 µM of Hoechst 33342 (ENZ-52401, Enzo Life Sciences) for 30 min at 37 °C. Cells expressing MitoSOX and Hoechst 33342 fluorescence were photomicrographed by a Keyence microscope and the fluorescent intensity was quantified using the Keyence software. Relative MitoSOX fluorescence intensity was determined as total MitoSOX fluorescence intensity divided by the number of Hoechst 33342-stained nucleus. An antioxidant NAC was utilized to assess MitoSOX™ staining specificity in cells treated with EtOH.

### 2.5. Quantitative Reverse-Transcription Polymerase Chain Reaction (qRT-PCR)

Total RNA was isolated using RNeasy Mini Kit (Qiagen, Valencia, CA, USA), and cDNA was synthesized using High Capacity cDNA Reverse Transcription kits (4368814, Thermo Fisher Scientific) according to the manufacturer’s instructions. qRT-PCR was performed with paired primers for *ATG7*-F (5′-AGCCCACAGATGGAGTAGCAGTTT-3′) and *ATG7*-R (5′-TCCCATGCCTCCTTTCTGGTTCTT-3′), *GAPDH*-F (5′-CCAGGTGGTCT CCTCTGACTTC-3′) and *GAPDH*-R (5′-GTGGTCGTTGAGGGCAATC-3′) using Power SYBR Green PCR Master Mix (Thermo Fisher Scientific). All reactions were carried out on the StepOnePlus Real-Time PCR System (Thermo Fisher Scientific). Relative *ATG7* level was normalized to *GAPDH* serving as an internal control gene.

### 2.6. Xenograft Transplantation Experiment

Xenograft transplantation experiments in immunodeficient mice were performed as described [19,27] under a protocol approved by the Institutional Animal Care and Use Committee. TE11-RFP and TE14-RFP cells propagated in monolayer culture were trypsinized and suspended in 50% Matrigel^®^ Basement Membrane Matrix and implanted subcutaneously into the dorsal flanks of 8-week-old athymic nu/nu mice (Taconic Biosciences, Hudson, NY, USA). EtOH at a concentration of 10% was given for 4–6 weeks in drinking water ad libitum, starting from time 0 or 2 weeks after tumor cell implantation. Tumor volume was weekly measured using a digital caliper and calculated using the formula of tumor volume (mm^3^) = [width (mm)]^2^ × length (mm) × 0.5. During EtOH treatment, 4MP (10 mg/kg; Sigma-Aldrich) or Dulbecco’s PBS (DPBS) (vehicle control) was intraperitoneally injected 3 times per week, starting from time 0 of tumor cell implantation. Alternatively, hydroxychloroquine sulfate (HCQ) (60 mg/kg; Spectrum Chemical, New Brunswick, NJ, USA), autophagy flux inhibitor, or DPBS (vehicle control) was intraperitoneally injected daily, starting from 2 weeks after tumor cell implantation as described [15]. Tumors were dissociated into single cells and analyzed by flow cytometry for CD44H and CD44L cells expressing RFP as described [13].

### 2.7. Statistical Analyses

Data were analyzed as indicated using GraphPad Prism 8.0 software. *p* < 0.05 was considered significant. The differences between two groups were analyzed by Student’s *t*-test.

## 3. Results

### 3.1. EtOH Increases the Organoid Formation Capability of SCC Cells

To investigate how SCC cells may respond to EtOH exposure in a near-physiological setting, we utilized the 3D organoid system. We established single-cell derived SCC organoids [23], treated with or without EtOH for four days (Figure 1).

We first utilized ESCC cell lines TE11 and TE14 to generate primary organoids. The average size of organoids reached 10,000–20,000 µm^2^ per structure by day seven and continued to grow exponentially in the absence of EtOH (Figure 2A). When EtOH was added into culture medium, starting from day seven, 1–2% (*v*/*v*), but not 0.5%, EtOH suppressed organoid growth and cell viability (Figure 2A,B). EtOH at a 2% concentration impaired cell viability more severely compared to 1% EtOH (Appendix A). We next evaluated cell proliferation by measuring EdU incorporation at day 11 and day 14. We found that a higher percentage of organoids treated with 1% EtOH were EdU-positive (Figure 2C), suggesting that the surviving organoids may contain a unique subset of cells with increased proliferation in the presence of EtOH despite overall growth inhibition and increased cell death (Figure 2A,B).

We next asked how EtOH exposure of primary organoids may influence secondary organoid formation. We dissociated EtOH-treated primary organoids into a single-cell suspension and measured their ability to form secondary organoids in the absence of EtOH (Figure 1). In addition to ESCC cell lines, we included two independent ESCC PDO lines, ESC2 and ESC3. SCC cells from EtOH-treated primary organoids displayed consistently higher secondary organoid formation rate (OFR) in the subsequent passage compared with those from EtOH-untreated primary organoids (Figure 3), suggesting that EtOH exposure may foster SCC cells with an increased organoid initiating capability. 

### 3.2. EtOH Enriches CD44H Cells within Primary SCC Organoids

We hypothesized that CD44H cells have a high organoid-formation capability and that EtOH increased CD44H cells to promote organoid formation. Indeed, we found that that EtOH exposure resulted in an increased percentage of CD44H cells within TE11 and TE14 organoids (Figure 4A), in both time- and dose-dependent manners (Figure 4B,C). CD44H cell enrichment was also observed in five independent PDOs representing both ESCC and HNSCC (Figure 4D).

We have further evaluated the function of FACS-purified CD44H and CD44L cells isolated from primary organoids treated with or without EtOH. CD44H cells showed higher OFR than CD44L cells (Appendix A), consistent with a premise that EtOH may increase a proliferative cell population within 3D organoids (Figure 2C). These findings suggest that CD44H cell enrichment within EtOH-treated primary organoids may account for the increased secondary OFR. Additionally, there was no difference in the secondary OFR when CD44H cells from EtOH-treated organoids were compared to CD44H cells from EtOH-untreated control organoids (Appendix A), suggesting that EtOH may increase the proportion of CD44H to CD44L cells within 3D organoids but may not necessarily stimulate division of CD44H cells.

### 3.3. CD44H Cell Enrichment Involves EtOH Oxidation and Oxidative Stress

Mitochondrial redox homeostasis has a critical role in an induction of CD44H cells under a variety of stressors including hypoxia and chemotherapy [15,16,19,23]. Normal esophageal epithelial cells (keratinocytes) metabolize EtOH via ADH1B to produce acetaldehyde, a highly reactive and toxic compound that induces mitochondrial dysfunction, mitochondrial superoxide, and apoptosis [10,28]. We hypothesized that EtOH oxidation in SCC organoids may contribute to CD44H cell enrichment. To evaluate the effect of EtOH metabolization on mitochondrial function in SCC cells, we treated EtOH-exposed SCC cells with the ADH inhibitor 4MP. Using the MitoSOX assay, we determined that EtOH exposure induces mitochondrial superoxide in TE11 and TE14 cells in monolayer culture. Further, 4MP treatment attenuated the EtOH-induced MitoSOX signal (Appendix A), implicating ADH-mediated EtOH oxidation in superoxide production. The antioxidant compound NAC also attenuated the EtOH-induced superoxide production, indicating that reactive oxygen species (ROS) also have a role in this process (Appendix A). Under these conditions, both 4MP and NAC prevented EtOH from inducing CD44H cells within primary 3D organoids (Figure 5), suggesting that ADH-mediated EtOH oxidation and mitochondrial oxidative stress may mediate CD44H cell enrichment.

### 3.4. EtOH-Induced Mitochondrial Dysfunction and Apoptosis Are Limited in CD44H Cells

We next explored if specific cell populations within primary 3D organoids are vulnerable to EtOH-induced oxidative stress and related mitochondrial dysfunction [10]. We performed flow cytometry to measure mitochondrial membrane potential (Δψ_m_) and mitochondrial mass simultaneously utilizing MitoTracker Deep Red (MTDR; Δψ_m_-sensitive) and MitoTracker Green (MTG; Δψ_m_-insensitive) dyes [13,15]. We found that a small subset (<3%) of SCC cells within 3D organoids harbored decreased Δψ_m_ (low MTDR, indicating loss of ∆Ψm) compared with mitochondrial mass (MTG) (Figure 6A,B), suggesting that there is a basal level of mitochondrial dysfunction in SCC organoids. This cell population was significantly increased in response to EtOH stimulation (Figure 6A,B). Moreover, mitochondrial dysfunction was predominantly found within CD44L cells and was significantly increased upon EtOH exposure (Figure 6C,D).

We suspected that CD44L cells are more susceptible to EtOH-induced cell death. We assessed apoptosis using flow cytometry for cells stained with Annexin V and propidium iodide (PI) concurrently and found that EtOH exposure induced both early (Annexin V-positive, PI-negative) and late (Annexin V-positive, PI-positive) apoptosis (Figure 7A,B). Notably, apoptosis was detected predominantly in CD44L cells within EtOH-exposed organoids (Figure 7C,D), suggesting that CD44H cells may be capable of negating EtOH-induced oxidative stress and apoptosis.

### 3.5. CD44H Cells Survive EtOH-Induced Oxidative Stress by Autophagy

Since autophagy is activated as a cytoprotective mechanism in SCC cells under stress conditions [15,16,19,23], we hypothesized that autophagy may protect CD44H cells from EtOH-induced oxidative stress and apoptosis. We stained cells with cyto-ID, an autophagy vesicle (AV)-identifying fluorescent dye to evaluate autophagy in SCC organoids. EtOH exposure increased AV content in TE11 and TE14 3D organoids and this effect was further augmented by concurrent treatment with chloroquine (CQ) to inhibit lysosome-mediated clearance of AVs (Figure 8A). Moreover, co-staining of 3D organoids for CD44 and cyto-ID revealed that CD44H cells had a higher AV content than CD44L cells (Figure 8B). We have further confirmed that EtOH increases AV content and that CD44H cells had a higher AV content within SCC PDOs (Figure 8C,D), except HSC1 where AV content was comparable between CD44L and CD44H cells (data not shown). 

We next assessed the functional consequences of autophagy inhibition. Autophagy flux inhibition with CQ increased the mitochondrial superoxide level in EtOH-treated TE11 and TE14 cells in monolayer culture (Appendix A), suggesting that autophagy may limit EtOH-induced oxidative stress. In 3D organoids, CQ augmented EtOH-induced apoptosis (Appendix A), resulting in a decreased secondary organoid formation upon subculture (Appendix A), suggesting that autophagy may contribute to CD44H cell enrichment by limiting oxidative stress and apoptosis. Indeed, either pharmacological autophagy flux inhibition by CQ or RNA interference directed against ATG7, a key regulator of AV assembly, suppressed CD44H cell enrichment in EtOH-treated TE11 and TE14 3D organoids (Figure 9, Appendix A). 

### 3.6. Alcohol Drinking Enriches Intratumoral CD44H Cells via Autophagy to Promote Tumor Growth

Finally, we evaluated the effect of alcohol consumption on SCC tumor growth and CD44H enrichment in mice exposed to EtOH. We subcutaneously transplanted TE11-RFP and TE14-RFP cells into the dorsal flanks of athymic nu/nu mice and supplemented their drinking water with 10% EtOH for ad libitum consumption. Four to six weeks of EtOH treatment increased tumor growth compared to vehicle control groups (Figure 10A,B, and Appendix A). Concurrent 4MP treatment started from the time of tumor cell implantation (day zero) prevented EtOH from stimulating tumor growth, implicating ADH-mediated EtOH oxidation in the acceleration of ESCC tumor growth (Figure 10A). Flow cytometry analysis of dissociated xenograft tumors indicated that intratumoral CD44H cells are enriched in mice fed with alcohol (Figure 10C and Appendix A). Importantly, autophagy flux inhibition by hydroxychloroquine (HCQ) suppressed TE14-RFP tumor growth and CD44H cell enrichment in mice fed with 10% EtOH (Figure 10A,C), indicating that autophagy is required for alcohol-induced tumor growth. In aggregate, these results suggest that EtOH promotes SCC tumor growth by fostering the intratumoral CD44H cell population.

## 4. Discussion

### 4.1. The 3D Organoid and Xenograft Models Shed Light upon the Role of EtOH in Tumor Biology

In this study, we utilized the 3D organoid culture and xenograft transplantation models to identify how HNSCC and ESCC cells respond to EtOH in vitro and in vivo. SCC cells metabolize EtOH, leading to mitochondrial superoxide production, mitochondrial depolarization, and apoptosis. However, a subpopulation of CD44H SCC cells survive EtOH-induced oxidative stress via autophagy, promoting enhanced tumor growth. Therefore, EtOH exposure not only causes cell injury but also permits the enrichment of a subset of SCC cells with high malignant potential.

The 3D organoid system serves as a physiologically relevant experimental platform to determine effects of epithelial exposure to harmful environmental chemicals such as alcohol and acetaldehyde [10,28] that are linked to the pathogenesis of HNSCC and ESCC as well as other alcohol-associated cancers [8]. We have recently demonstrated that normal nontransformed (immortalized) human esophageal epithelial cells undergo cell-cycle arrest or apoptosis coupled with mitochondrial dysfunction in response to EtOH exposure [10]. This study indicates that the majority of heterogeneous SCC cells have similar responses to EtOH as normal cells. However, the presence of CD44H CSCs in SCCs enable these tumors to grow despite the deleterious effects of EtOH exposure. Future studies will address whether EtOH exposure in normal cells results in CD44H cell conversion, which would represent a key step in tumorigenesis. 

### 4.2. 3D Organoids Reveal HNSCC and ESCC CSCs Homeostasis under EtOH Exposure

Earlier studies have explored the effect of EtOH upon generation of CSCs (see Introduction section) in several tumor types. EtOH induces CD133/Nanog-positive liver CSCs through synergism between hepatitis C viral protein and the Toll-like receptor 4 (TLR4)-mediated signaling [30]. The erbB2-p38gamma MAPK pathway has a vital role in induction of breast CSCs upon EtOH exposure in the MMTV-neu transgenic mouse model and breast cancer cell line MCF7 [31]. Chronic EtOH exposure transforms normal human pancreatic ductal epithelial cells to induce CSCs with high CD44 expression [32]. In this study, we characterized CSCs in established tumors, but not malignant transformation or the CSC generation under chronic EtOH exposure. However, to our knowledge, this is the first study to demonstrate that EtOH influences the homeostasis of proliferative human HNSCC/ESCC CD44H CSCs. Cells from EtOH-treated primary organoids exhibited higher secondary OFR in a consistent manner for all cell lines and PDOs tested (Figure 3), which we attribute to EtOH-mediated CD44H enrichment (Figure 4). Supporting these findings, CD44H cells had a higher OFR than CD44L cells (Appendix A). However, the extent of CD44H cell induction (~4-fold; Figure 4) was generally greater than that for secondary OFR (1.3-to 1.5-fold; Figure 3). Thus, it is possible that EtOH-induced high organoid-formation capability may represent a subset of, but not all, CD44H cells. Future studies will address this possibility by evaluating additional CSC markers such as aldehyde dehydrogenase. However, CSCs are heterogeneous and currently available CSC markers are not as well defined as CD44. More work is needed to develop a comprehensive understanding of the different subpopulations of HNSCC/ESCC CSCs.

### 4.3. Limitations of the 3D Organoid Model to Study Cancer Cell Response to EtOH

Although the 3D organoid system has been proven to be highly reliable as a tool to recapitulate structures and functions of original tissues and predict therapeutic response [24,33,34], there are several limitations to this model. Whether mucosal SCC lesions are continuously exposed to EtOH for a period of four days or longer at the concentrations utilized in this study is unknown. Alcohol admixed with saliva is detectable in oral fluid for up to 24 hours after consumption [35] and it is not unrealistic for heavy drinkers to have 1–2% EtOH on the esophageal mucosal surface during and after consumption of alcohol beverages with a high (>10–40%) EtOH content. As normal epithelial cells tolerate 20% EtOH for 15 s, the standard time for a swallowed liquid to pass through the human esophageal lumen [36,37,38], future experimental conditions should include intermittent exposures to high concentrations of EtOH. 

Additionally, while the 3D organoid system offers insights into the effects of EtOH exposure on cancer cells, alcohol may influence SCC cells in a non-cell autonomous manner by modifying the tumor microenvironment. EtOH promoted TE11 and TE14 xenograft tumor growth (Figure 10). By contrast, EtOH suppressed TE11 and TE14 organoid growth (Figure 1), suggesting that the influence of EtOH on tumor biology is more complex in vivo beyond differential levels of EtOH and its metabolites to which SCC cells are exposed. For example, alcohol may promote tumor growth by activating tumor angiogenesis [39]. In the tumor microenvironment, hypoxia regulates CD44 expression [40,41] and CD44H cell homeostasis [42]. Therefore, hypoxia may have an additive or synergic role with EtOH and its metabolites in promoting CD44H cell enrichment. Although our 3D organoids grew under normoxic (21% O_2_) conditions, oxygen tension of the inner cell mass of growing organoids is unknown. Future studies should address the potential functional interplay between hypoxia and EtOH in organoids as well as xenograft tumors following genetic or pharmacological modification of the hypoxia pathway [43,44].

Alcohol may also promote tumor growth by suppressing the tumor immune microenvironment [45]. Although immunodeficient athymic nude mice were utilized, our data from xenograft transplantation models do not exclude the potential effects of alcohol drinking upon residual immune cells that may potentially limit tumor growth in alcohol-unfed control mice. To address the influence of alcohol upon tumor immunity, SCC organoids can be generated from genetically engineered mice for allograft transplantation experiments in immunocompetent syngeneic mice. This study is underway in our laboratory. Alcohol may also induce fibrosis, a feature of liver cirrhosis that fosters alcohol-related hepatocellular carcinoma [46]. Potential contribution of these factors may be addressed by coculture of stromal cells and 3D organoids as demonstrated with pancreatic CSCs [47]. Finally, alcohol may promote tumor growth by altering the hormonal environment in vivo, as implicated in breast cancer where alcohol elevates circulating estrogen levels [48].

### 4.4. Alcohol Metabolism, Mitochondrial Oxidative Stress, and Autophagy in SCC Cells

This study is the first to demonstrate that SCC cells can oxidize EtOH via ADH (Figure 5). Other enzymes, such as CYP2E1, are also implicated in this process. Although the role of CYP2E1 in EtOH metabolism was not addressed in this study, RNA interference experiments suggested that ADH may have a greater contribution to EtOH oxidation than CYP2E1 in esophageal epithelial cells [10]. Future studies should clarify involvement of these enzymes through targeted modifications in SCC cells, especially in xenograft models, to evaluate to what extent SCC cells may oxidize EtOH in circulation.

This study also revealed that EtOH exposure causes mitochondrial damage, which results in superoxide production, oxidative stress, and apoptosis in non-CD44H cells. EtOH-induced oxidative stress and apoptosis may be facilitated by acetaldehyde whose clearance is regulated by ALDH2. Heterozygous or homozygous single-nucleotide polymorphism (SNP) of ALDH2 (ALDH2*2) [49] is carried by >8% of the world’s population, and decreases its catalytic activity compared with wild-type ALDH2 (ALDH2*1) [50,51]. In this study, we determined that the ESCC cell lines TE11 and TE14 have heterozygous (ALDH2*1/ALDH2*2) ALDH2 alleles while all PDOs (ESC2, ESC3, HSC1-3) carry homozygous (ALDH2*1/ALDH2*1) wild-type ALDH2 alleles (Appendix A). No correlation was noted between the ALDH2 status and the extent of EtOH-induced CD44H cell enrichment (Figure 4). Other genetic factors (e.g., SNP in ADH and CYP2E1) may influence CD44H cell homeostasis. Given genetic heterogeneity in individual cell lines and PDOs, CRISPR-Cas9-mediated alterations of ALDH2*1 and ALDH2*2 will better delineate the role of ALDH2 SNP in the syngeneic background. Creation of such PDO lines is underway in our laboratory. In Aldh2-deficient murine esophageal epithelial cells, delayed acetaldehyde clearance resulted in mitochondrial superoxide-mediated oxidative stress and cell death that was augmented by inhibition of autophagy [28]. Thus, autophagy appears to serve as a common mechanism for both normal epithelial cells and SCC cells to cope with oxidative stress associated with alcohol metabolism. This finding may have translational potential, as pharmacological inhibition of autophagy flux by CQ appeared to prevent EtOH from inducing CD44H cells (Figure 9 and Figure 10).

EtOH-induced oxidative stress may lead to activation of cell signaling pathways that regulate autophagy. In normal cells, EtOH exposure results in reduced mammalian targets of rapamycin complex 1 (mTORC1) signaling, a key repressor of autophagy [10]. Consistent with these data, we determined that EtOH treatment resulted in decreased phosphorylation of mTORC1 substrates in TE14 cells (Appendix A). Future studies will characterize the effect of EtOH exposure on mTORC1 signaling, especially in CSCs. 

Despite accumulation of autophagosomes and the inhibitory effect of autophagy flux upon EtOH-induced CD44H cell enrichment (Figure 8, Figure 9, Figure 10), changes in expression of autophagy regulators p62 sequestosome 1 (SQSTM1) and microtubule-associated protein 1A/1B-light chain 3 (LC3) proteins were not detected by immunoblot analysis (data not shown). This result is potentially due to autophagy activation occurring only in a limited number of cells that display EtOH-induced mitochondrial depolarization and apoptosis (Figure 6 and Figure 7).

In addition to autophagy, other cytoprotective mechanisms may have a role in CD44H cell enrichment. In HNSCC and ESCC cells, mitochondrial superoxide dismutase 2 (SOD2) mediates CD44H cell induction coupled with autophagy [15] as well as epithelial-mesenchymal transition [16]. Interestingly, CD44-mediated signaling regulates glycolysis as well as antioxidant-reduced glutathione to promote tumor growth and therapy resistance [52,53]. Additionally, CD44-mediated signaling activates nuclear factor NRF2, a key regulator of antioxidant genes to regulate CD44H breast CSCs [54]. Therefore, CD44 may play a central role in the redox homeostasis under alcohol-induced stress and other stress conditions such as chemotherapy in SCC cells [23].

## 5. Conclusions

This study provides mechanistic insights describing how EtOH metabolism may influence both CSC and non-CSC subpopulations of HNSCC and ESCC tumors and organoids. HNSCC and ESCC cells oxidize alcohol to produce toxic metabolites that result in mitochondrial damage and apoptosis. Non-CSC subpopulations of HSNCC and ESCC cells do not tolerate alcohol injury, as damaged mitochondria accumulate and these cells undergo apoptosis. However, existing CSC subpopulations of HNSCC and ESCC organoids are resistant to alcohol injury; these cells can dampen the deleterious effects of EtOH exposure through the autophagy-mediated clearance of damaged mitochondria. These cells are therefore able to form organoids at a higher rate and are associated with increased xenograft tumor growth following EtOH exposure.

These findings may be clinically relevant. Given high tumorigenic potential of CD44H cells, SCC patients should abstain from drinking alcohol to minimize the chance of post-therapeutic recurrence. Additionally, since autophagy has an essential role in regulating redox balance in SCC cells and contributes to the survival and enrichment of CD44H cells under EtOH-induced oxidative stress, pharmacological autophagy inhibition may benefit SCC patients with a history of heavy alcohol consumption. Finally, PDOs may serve as an excellent platform to assess individual EtOH metabolism capability as well as to predict the effect of autophagy inhibition in translational applications for personalized medicine.

## Figures and Tables

**Figure 1 biomolecules-11-01479-f001:**
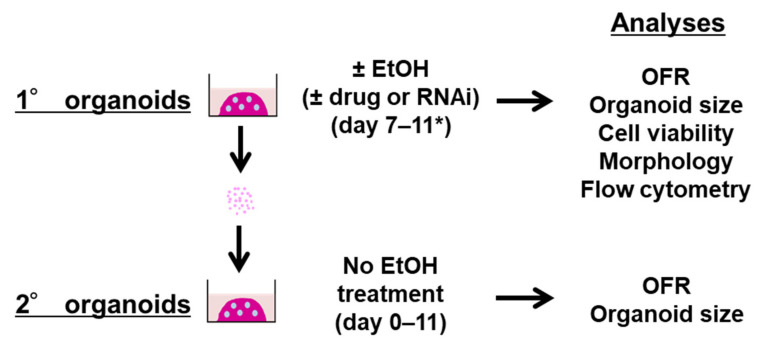
Experimental design. Primary (1°) SCC organoids were established and treated with or without EtOH for 4 days, starting from day 7 in the presence or absence of ADH inhibitor 4-methylpyrazole (4MP), autophagy flux inhibitor chloroquine (CQ), or doxycycline (DOX) to induce shRNA directed against *ATG7*. Cells dissociated from 1° SCC organoids were passaged to grow secondary (2°) SCC organoids in subculture. Organoid formation rate (OFR) and organoid growth (size) were determined for both 1 and 2° SCC organoids. 1° SCC organoids were also analyzed for cell viability, morphology (H&E staining) as well as flow cytometry to determine cell surface CD44 expression, proliferation (EdU incorporation), apoptosis (Annexin V staining), mitochondrial mass (MTG) and membrane potential (MTDR), and autophagy (cyto-ID). * Treatment was extended up to day 14 in experiments shown in Figure 2C.

**Figure 2 biomolecules-11-01479-f002:**
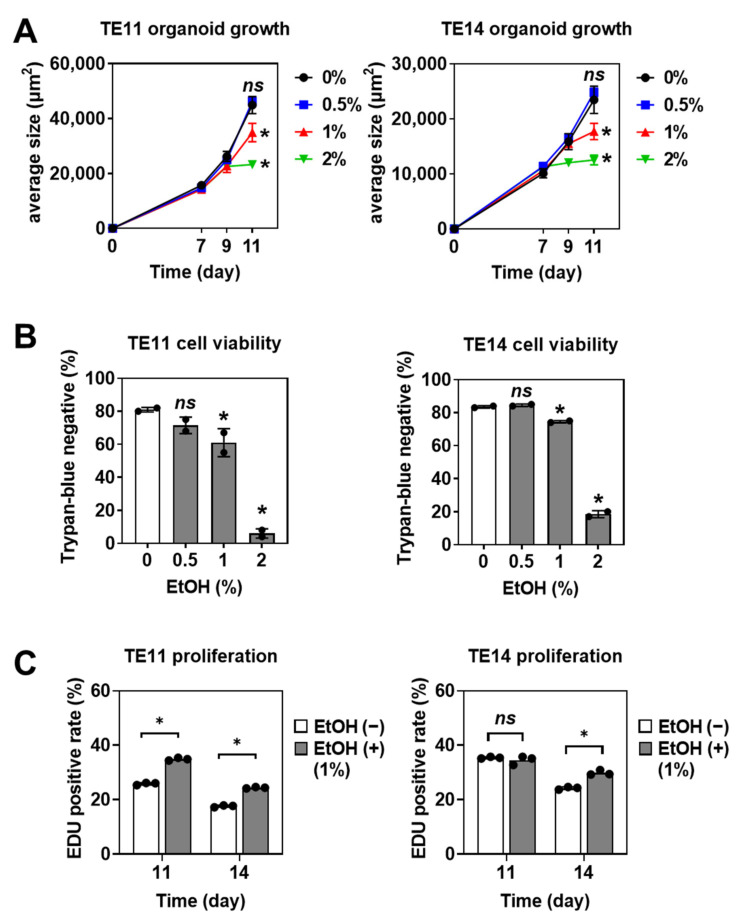
EtOH suppresses organoid growth and cell viability while permitting proliferation of a subset of cells within the 1° SCC organoids. (**A**,**B**) Organoids generated from TE11 and TE14 cells were treated with EtOH at indicated concentrations for 4 days, starting from day 7. Average organoid sizes of independent wells were plotted in (**A**). Cell viability at day 11 was determined by trypan-blue exclusion test in (**B**). (**C**) TE11 and TE14 organoids were treated with or without 1% EtOH for 4 days and 7 days and harvested at indicated time points. Organoids were exposed to EdU for 2 h prior to harvest. Cell proliferation was assessed as EdU uptake determined by flow cytometry. ns, not significant vs. 0% EtOH; *, *p* < 0.05 vs. 0% EtOH, *n* = 3.

**Figure 3 biomolecules-11-01479-f003:**
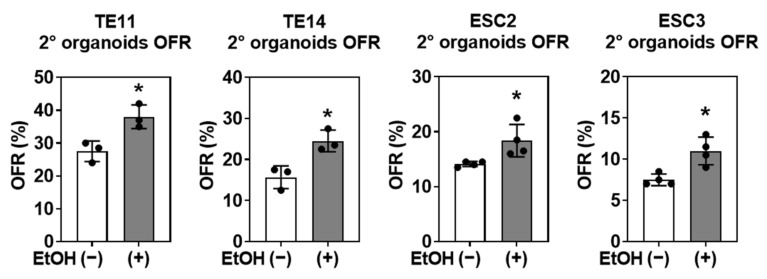
SCC cells isolated from EtOH-exposed primary organoids showed increased secondary (2°) organoid formation. TE11 and TE14 organoids and indicated PDOs were treated with or without 1% EtOH for 4 days in primary organoids. Secondary (2°) OFR in subculture was determined and plotted in bar graphs. *, *p* < 0.05 vs. EtOH (−), *n* = 3.

**Figure 4 biomolecules-11-01479-f004:**
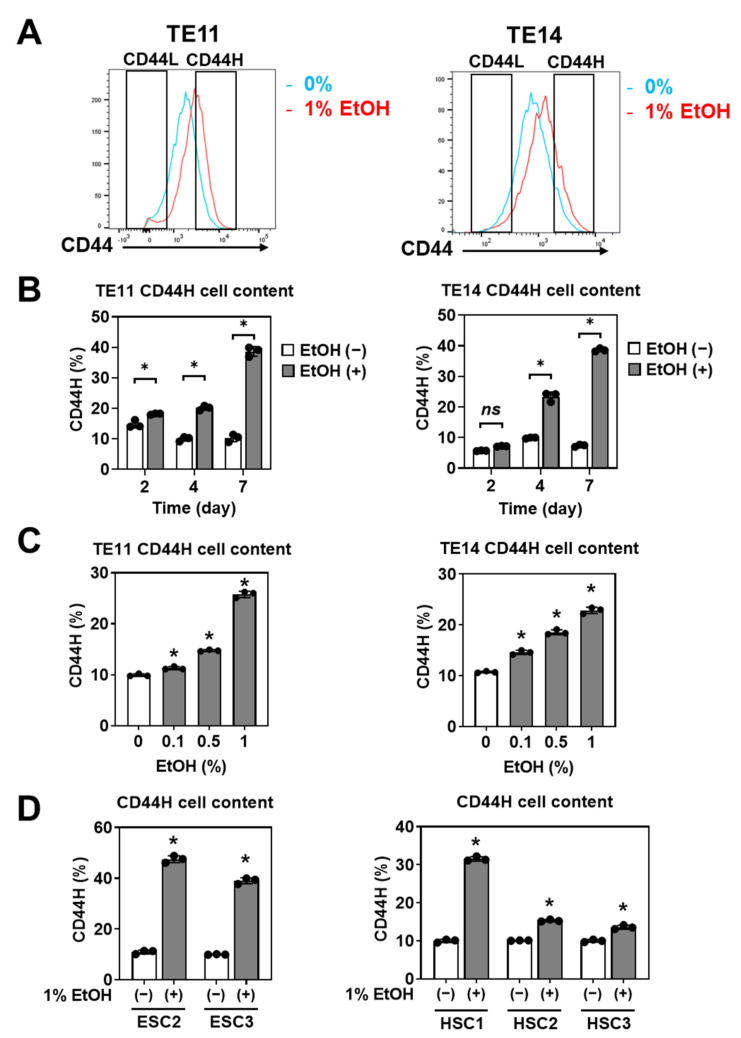
EtOH increases CD44H cells within 1° SCC organoids. (**A**,**B**) TE11 and TE14 organoids were treated with or without 1% EtOH for 2, 4 or 7 days. Dissociated organoid cells were analyzed by flow cytometry to determine the CD44 expression levels. CD44H and CD44L cells were identified as the top 10% and the bottom 10% of CD44 expressing cells in control organoids grown for 11 days without EtOH. Representative histogram plots are shown for organoids treated with EtOH for 4 days in (**A**). CD44H cell content was determined at each time point in (**B**). (**C**) Organoids generated with ESCC cell lines (TE11 and TE14) were treated with indicated concentrations of EtOH for 4 days to determine CD44H cell content. (**D**) PDO lines (ESC2, ESC3, and HSC1-3) were treated with 1% EtOH for 4 days to determine CD44H cell content. ns, not significant vs. EtOH (−); * *p* < 0.05 vs. EtOH (−) or 0% EtOH. *n* = 3 in (**B**–**D**).

**Figure 5 biomolecules-11-01479-f005:**
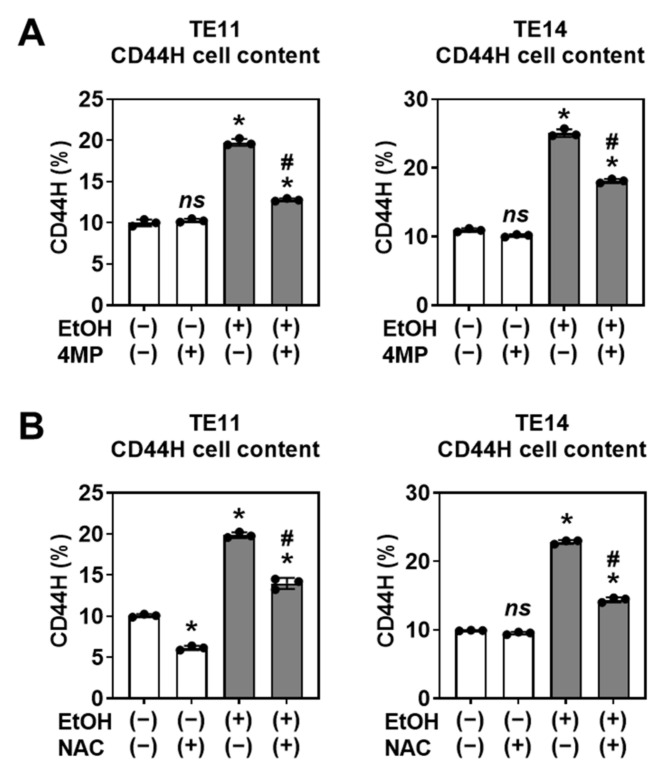
CD44H cell enrichment involves ADH-mediated EtOH oxidation and oxidative stress. TE11 and TE14 organoids were treated with or without 1% EtOH for 4 days along with or without 2 mM of 4MP (**A**) or 10 mM of NAC (**B**). Dissociated organoid cells were analyzed by flow cytometry to determine the CD44H cell contents. ns, not significant vs. EtOH (−) and 4MP (−) or EtOH (−) and NAC (−); * *p* < 0.05 vs. EtOH (−) and 4MP (−) or EtOH (−) and NAC (−); ^#^
*p* < 0.05 vs. EtOH (+) and 4MP (−) or EtOH (+) and NAC (−), *n* = 3.

**Figure 6 biomolecules-11-01479-f006:**
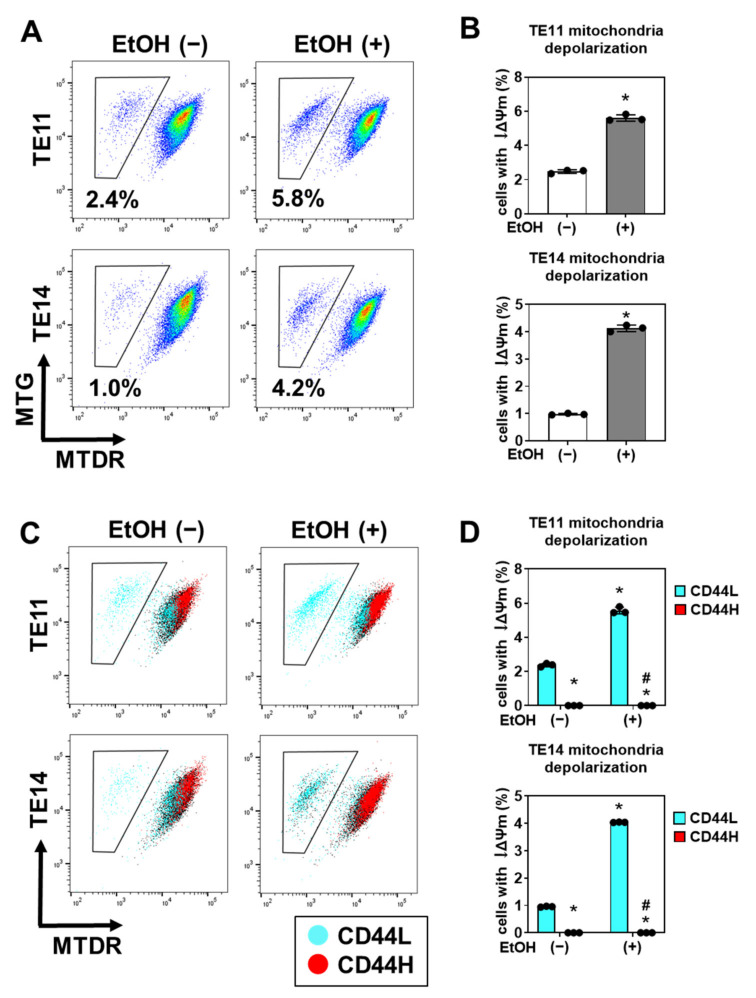
EtOH induces mitochondrial depolarization in CD44L cells within 1° SCC organoids. TE11 and TE14 organoids were treated with or without 1% EtOH for 4 days. (**A**,**B**) Dissociated organoid cells were analyzed by flow cytometry to determine mitochondrial mass (MTG) and mitochondrial depolarization (MTDR). * *p* < 0.05 vs. EtOH (−). Representative dot plots are shown in (**A**). Bar graphs display quantitative representation of cells with mitochondria depolarization (i.e., decreased MTDR staining) in (**B**). (**C**,**D**) Dissociated organoid cells were co-stained for CD44, MTG and MTDR to determine mitochondrial mass and mitochondrial depolarization in CD44H or CD44L cells within organoids. Representative dot plots are shown in (**C**). Bar graphs display quantitative representation of cells with mitochondria depolarization in (**D**). * *p* < 0.05 vs. CD44L in EtOH (−); ^#^
*p* < 0.05 vs. CD44L in EtOH (+), *n* = 3.

**Figure 7 biomolecules-11-01479-f007:**
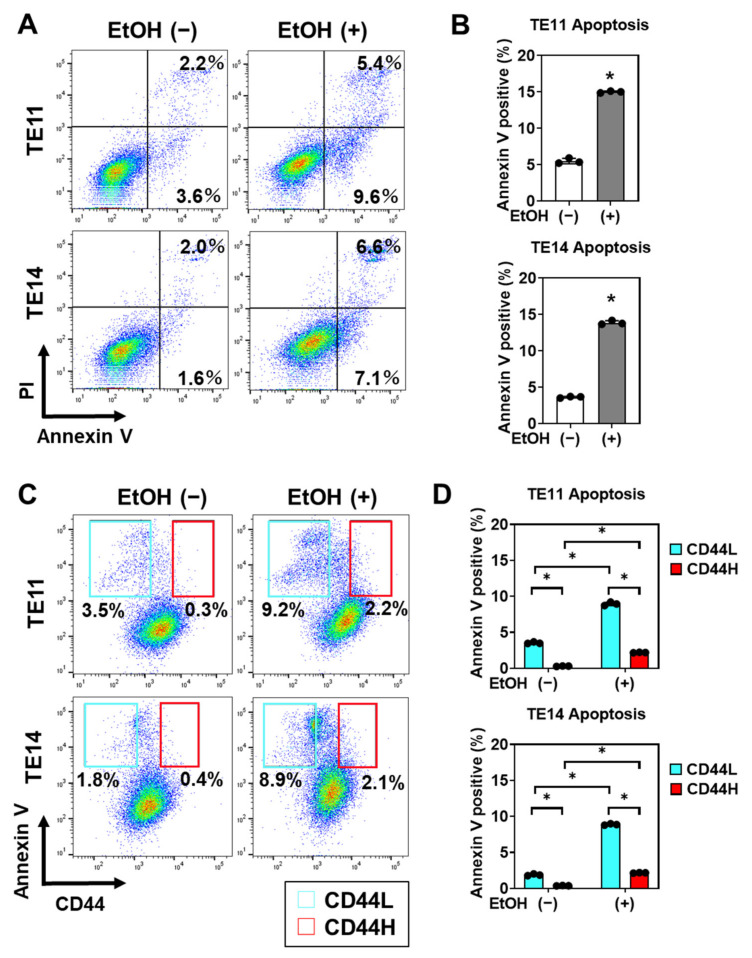
EtOH induces apoptosis in CD44L cells within 1° SCC organoids. TE11 and TE14 organoids were treated with or without 1% EtOH for 4 days. (**A**,**B**) Dissociated organoid cells were co-stained with PI and Annexin V, and analyzed by flow cytometry to determine the apoptotic cell population represented by Annexin V-positive cells. Representative dot plots are shown in (**A**). Bar graphs show quantitative representation of Annexin V-positive apoptotic cells in (**B**). (**C**,**D**) Dissociated organoid cells were stained with Annexin V along with CD44, and subjected to flow cytometry analysis to determine apoptosis in CD44H or CD44L cells. Representative dot plots are shown in (**C**). Bar graphs show quantitative representation of Annexin V-positive apoptotic cells in CD44L and CD44H cell fractions (**D**). * *p* < 0.05 vs. EtOH (−), *n* = 3.

**Figure 8 biomolecules-11-01479-f008:**
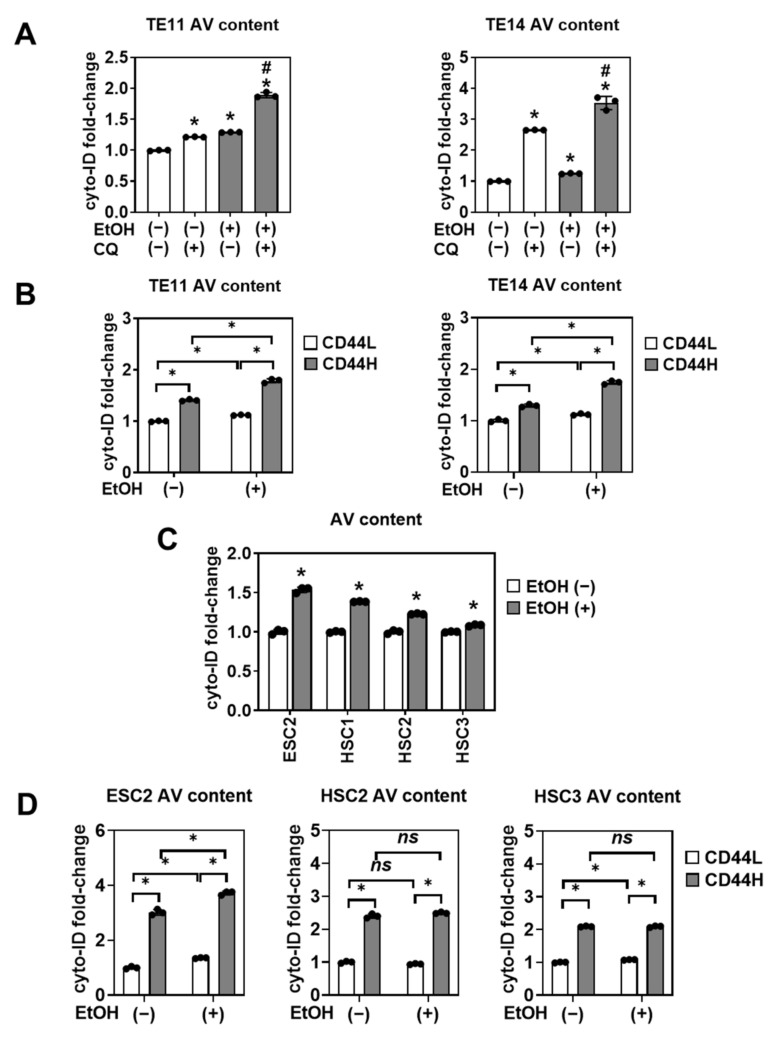
EtOH induces autophagy in 1° SCC organoids. (**A**,**C**) TE11 and TE14 organoids (**A**) and PDOs (**C**) were treated with or without 1% EtOH for 4 days along with or without 2 µM of CQ. Dissociated organoid cells were analyzed by flow cytometry to determine the AV contents. * *p* < 0.05 vs. EtOH (−) and CQ (−); ^#^
*p* < 0.05 vs. EtOH (+) and CQ (−), *n* = 3 in (**A**). * *p* < 0.05 vs. EtOH (−), *n* = 3 in (**C**). (**B**,**D**) Co-staining of CD44 and cyto-ID was performed to measure the AV contents in CD44H and CD44L cells. ns, not significant; * *p* < 0.05, *n* = 3.

**Figure 9 biomolecules-11-01479-f009:**
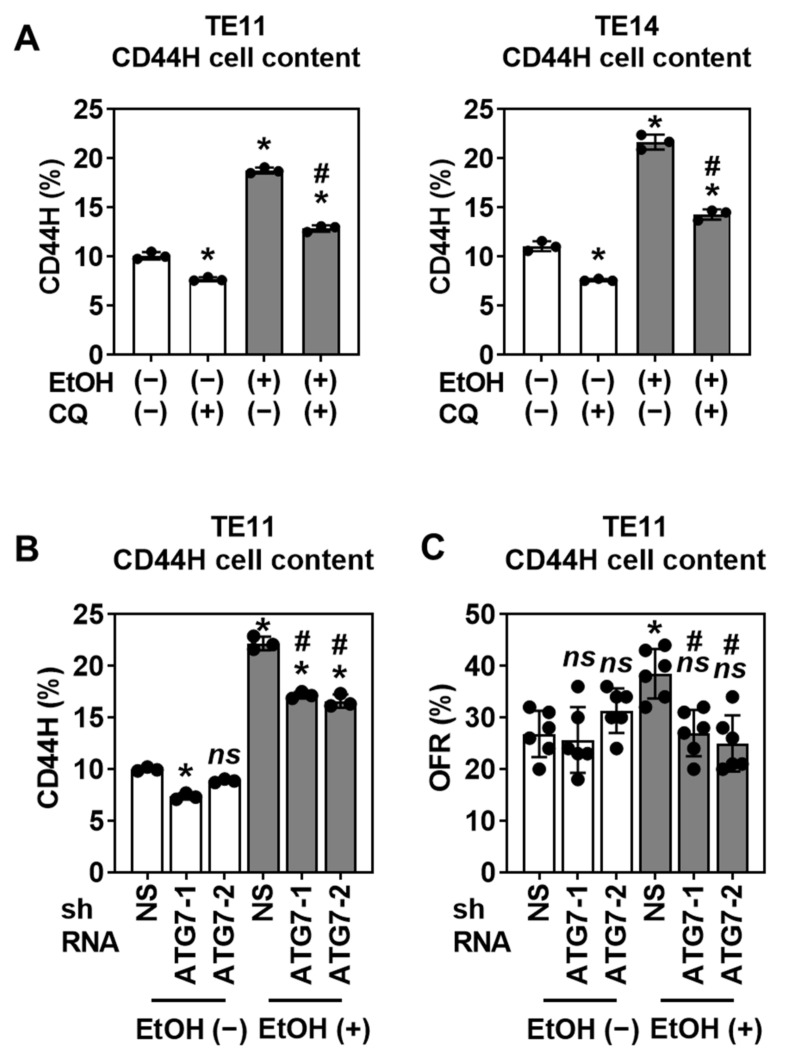
Autophagy mediates CD44H cell enrichment within EtOH-exposed 1° SCC organoids. (**A**) TE11 and TE14 organoids were treated with or without 1% EtOH for 4 days along with or without 2 µM of CQ. Dissociated organoids were analyzed by flow cytometry for CD44H cell contents. * *p* < 0.05 vs. EtOH (−) and CQ (−); ^#^
*p* < 0.05 vs. EtOH (+) and CQ (−), *n* = 3. (**B**) TE11 organoids of indicated genotypes were treated with or without 1% EtOH for 4 days along with DOX to induce shRNA. Note that DOX-untreated cells with shRNA had no impact upon ATG7 expression (Appendix A). Dissociated organoid cells were analyzed by flow cytometry to determine the CD44H cell contents. ns, not significant vs. EtOH (−) and NS shRNA (i.e., nonsilencing control); * *p* < 0.05 vs. EtOH (−) and NS shRNA; ^#^
*p* < 0.05 vs. EtOH (+) and NS shRNA, *n* = 3. (**C**) TE11 organoids of indicated genotypes were treated with or without 1% EtOH for 4 days along with DOX to induce shRNA in 1° organoids. Organoids were passaged to grow 2° organoids in subculture in the absence of DOX. OFRs of 2° organoids were determined and plotted in bar graphs. ns, not significant vs. EtOH (−) and NS shRNA; * *p* < 0.05 vs. EtOH (−) and NS shRNA; ^#^
*p* < 0.05 vs. EtOH (+) and NS shRNA, *n* = 6.

**Figure 10 biomolecules-11-01479-f010:**
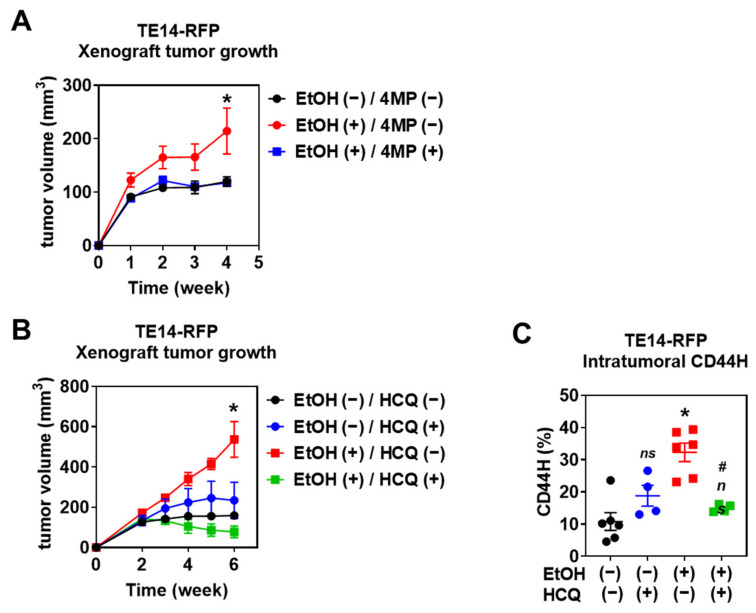
Autophagy mediates CD44H cell enrichment within xenograft tumors transplanted in alcohol-fed immunodeficient mice. TE14-RFP cells were subcutaneously injected to the lower back of immunodeficient mice. (**A**) Mice were given 10% EtOH in drinking water along with or without ADH inhibitor 4MP for 6 weeks, starting from the day when tumor cells were implanted in indicated three groups (*n* = 6/group). Tumor volume was measured once a week and plotted in graphs. * *p* < 0.05 vs. EtOH (−) and 4MP (−); or EtOH (+) and 4MP (+). (**B**,**C**) Mice were given 10% EtOH in drinking water along with or without 60 mg/kg/day HCQ for 4 weeks, starting 2 weeks after tumor cell implantation in indicated four groups (*n* = 16/group), and sacrificed at the 6-week time point. Tumor volume was measured once a week and plotted in graphs. * *p* < 0.05 vs. EtOH (−) and HCQ (−); or EtOH (+) and HCQ (+). (**C**) Harvested tumors were dissociated and analyzed by flow cytometry to determine intratumoral CD44H cells. ns, not significant vs. EtOH (−) and HCQ (−); * *p* < 0.05 vs. EtOH (−) and HCQ (−); ^#^
*p* < 0.05 vs. EtOH (+) and HCQ (−). *n* = 6 for EtOH (−) and HCQ (−), *n* = 6 for EtOH (+) and HCQ (−), *n* = 4 for EtOH (−) and HCQ (+), and *n* = 4 for EtOH (+) and HCQ (+).

## Data Availability

Not applicable.

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
