# Peer review of "Alcohol Metabolism Enriches Squamous Cell Carcinoma Cancer Stem Cells That Survive Oxidative Stress via Autophagy"

_biomolecules, 2021, doi:10.3390/biom11101479_

Round 1

Reviewer 1 Report

This is a study showing the enrichment of CD44-high cells in cell lines and patient derived organoids upon the exposure to 1% EtOH.  They further the enrichment of such cells were involved in the ATG7-mediate autophagy and ROS production.  Collectively, more future studies will definitely be needed to clarify the detailed mechanism responsible for the observations.

  1. Although the authors stated that these studies are not focused in studying alcohol metabolism, it is still not a bad idea to clarify the genotype of ALDH2, a key enzyme involved in the accumulation of toxic acetaldehyde after alcohol metabolism, in the tested cell lines and patient-derived organoids. Plus it is known that ALDH2-2* variant can contribute to the alcohol toxicity.  What would be the genotype for all the cell lines and patient-derived organoids used in this study? They might want to provide such information and include some discussion in the report.
  2. Since hypoxia was previously shown to induce CD44 expression, and the cells in the interior of organoids could potentially experience hypoxia during the 3-dimensional growth, it will be interesting to examine if the hypoxia plays any additive or synergic role with the exposure of EtOH and its metabolite in promoting the expression of CD44 by using fluorescence microscopy.

Author Response

Reviewer 1

“This is a study showing the enrichment of CD44-high cells in cell lines and patient derived organoids upon the exposure to 1% EtOH. They further the enrichment of such cells were involved in the ATG7-mediate autophagy and ROS production.  Collectively, more future studies will definitely be needed to clarify the detailed mechanism responsible for the observations.”

We agree and have extended discussion about the roles of ALHD2 and hypoxia upon EtOH-induced enrichment of esophageal cancer cells with CD44 high (CD44H) expression.

  1. “Although the authors stated that these studies are not focused in studying alcohol metabolism, it is still not a bad idea to clarify the genotype of ALDH2, a key enzyme involved in the accumulation of toxic acetaldehyde after alcohol metabolism, in the tested cell lines and patient-derived organoids. Plus it is known that ALDH2-2* variant can contribute to the alcohol toxicity.  What would be the genotype for all the cell lines and patient-derived organoids used in this study? They might want to provide such information and include some discussion in the report.”

We agree that EtOH-induced cell toxicity involves multiple alcohol metabolizing enzymes including alcohol dehydrogenase, CYP2E1, and ALDH2. As shown in new Supplementary Table S1, we have now determined ALDH2 genotype for esophageal squamous cell carcinoma cell lines TE11 and TE14 as well as patient-derived organoids utilized in this study. Both esophageal cancer cell lines appeared to have a single nucleotide polymorphic (SNP) mutant allele (ALDH2*2). All patient-derived organoids were found to carry wild-type ALDH2 alleles (ALDH2*1) only. In this study, the presence or absence of ALDH2*2 alleles did not appear influence CD44H cell induction upon EtOH exposure (Figure 4D). Due to genetic heterogeneity and a limited sample size, we could not clarify the role of ALDH2 SNP in this study. We have discussed the development of engineered cell lines and organoid lines with altered ALDH2 SNP alleles in the syngeneic background to better delineate the role of ALDH2 SNP in CD44H cell homeostasis in EtOH-exposed cells (page 15, lines 686-697).

  1. “Since hypoxia was previously shown to induce CD44 expression, and the cells in the interior of organoids could potentially experience hypoxia during the 3-dimensional growth, it will be interesting to examine if the hypoxia plays any additive or synergic role with the exposure of EtOH and its metabolite in promoting the expression of CD44 by using fluorescence microscopy.”

Although exploring the potential functional interplays between hypoxia and EtOH metabolic pathways is outside of the scope of this manuscript, we have added this point into discussion (page 15, lines 655-661).

Reviewer 2 Report

In this manuscript, Shimonosono et al. revealed the mechanistic insight of alcohol-induced enrichment of squamous cell carcinoma cancer stem cells. Using 3D organoids generated from SCC cell lines, patient-derived xenograft tumors and patient biopsies, they showed that EtOH mitigates oxidative stress and apoptosis in SCC cells within organoids, but cancer stem cells (CSC) with high CD44 expression are resistant to EtOH-induced mitochondrial dysfunction and apoptosis by upregulating autophagy. Inhibition of autophagy augmented EtOH-mediated apoptosis of CD44H cells as well as reduced CD44H cell enrichment, xenograft tumor growth, and organoid formation rate.

The study is very well presented with the novel finding using the new organoid models with ESSC patient-derived cells and HNSCC patient-derived xenograft tumor models. The experiments are appropriately designed as well as interpretation of results and discussion are properly presented. I have few minor suggestions to improve the presentation of this study.

  1. Figure 4C, D legends are confusing. Figure 4D needs separate explanation in legend.
  2. Figure 7 legend was not properly explained. Legends for A, B and C, D should be separately explained.
  3. In method and result section, the authors noted that mice with tumor were treated with EtOH for 4 weeks. However, in Figure 10A legend, it is noted that mice were treated with EtOH for 6 weeks. Please verify it.
  4. Please define CSC when it is first time used in section 4.2.
  5. They should perform Western Blots to examine the levels of p62, LCII/I to support the autophagic induction in CD44H cells after EtOH treatment.
  6. Is there any differential regulation of mTOR pathway in SCC and C44H cells after EtOH treatment? At least they can discuss it as a limitation of this study.

Author Response

Reviewer 2

“In this manuscript, Shimonosono et al. revealed the mechanistic insight of alcohol-induced enrichment of squamous cell carcinoma cancer stem cells. Using 3D organoids generated from SCC cell lines, patient-derived xenograft tumors and patient biopsies, they showed that EtOH mitigates oxidative stress and apoptosis in SCC cells within organoids, but cancer stem cells (CSC) with high CD44 expression are resistant to EtOH-induced mitochondrial dysfunction and apoptosis by upregulating autophagy. Inhibition of autophagy augmented EtOH-mediated apoptosis of CD44H cells as well as reduced CD44H cell enrichment, xenograft tumor growth, and organoid formation rate. The study is very well presented with the novel finding using the new organoid models with ESSC patient-derived cells and HNSCC patient-derived xenograft tumor models. The experiments are appropriately designed as well as interpretation of results and discussion are properly presented. I have few minor suggestions to improve the presentation of this study.”

  1. “Figure 4C, D legends are confusing. Figure 4D needs separate explanation in legend.”

We agree. We have edited the legends for Figure 4C and 4D, as well as those for Figure 6.

  1. “Figure 7 legend was not properly explained. Legends for A, B and C, D should be separately explained.”

Thank you for picking this error. We agree and have edited the legend for Figure 7.

  1. “In method and result section, the authors noted that mice with tumor were treated with EtOH for 4 weeks. However, in Figure 10A legend, it is noted that mice were treated with EtOH for 6 weeks. Please verify it.”

We are sorry for this confusion. There were differential experimental conditions utilized. In Figure 10A, mice received EtOH for 4 weeks, starting from day 0 of tumor cell implantation. In Figure 10B and C, and Supplementary Figure 6, mice received EtOH for 4-6 weeks, starting at 2 weeks after tumor cell implantation. We have revised text (page 12, lines 554-557; page 13, lines 584, 587) and figure legends to clarify these points.

  1. “Please define CSC when it is first time used in section 4.2.”

We have defined cancer stem cells (CSCs) in the introduction section as cells with high expression of cell-surface CD44 (CD44H) glycoprotein that display increased malignant properties including invasion, metastasis, and therapy resistance in addition to a high tumor initiation capability. We have edited section 4.2 to direct readers to the Introduction section (page 14, lines 615-616).

  1. “They should perform Western Blots to examine the levels of p62, LCII/I to support the autophagic induction in CD44H cells after EtOH treatment.”

Despite accumulation of autophagosomes detected by flow cytometry for cyto-ID-stained cells and the inhibitory effect of autophagy flux (Figures 8-10), immunoblot analysis failed to detect changes in expression of autophagy regulators p62 sequestosome 1 (SQSTM1) and microtubule-associated protein 1A/1B-light chain 3 (LC3) proteins in squamous cell carcinoma cells upon EtOH exposure (data not shown). Given a relatively small fraction of cells showing EtOH-induced mitochondrial depolarization (Figure 6) and apoptosis (Figure 7), we reasoned that autophagy may be occurring in a limited number of cells only to be detected by immunoblotting on whole cell lysates. We have discussed accordingly (page 16, lines 752-758).

  1. “Is there any differential regulation of mTOR pathway in SCC and C44H cells after EtOH treatment? At least they can discuss it as a limitation of this study.”

Our new Western blotting data (Supplementary Fig. S7) suggests that EtOH may suppress phosphorylation of mTORC1 substrates in squamous cell carcinoma cells. Although we could not determine the mTORC1 activity within CD44H cell subpopulation in this study, we suspect that mTORC1 may have a role in regulation of CD44H cell homeostasis under EtOH-induced oxidative stress as discussed (page 16, lines 745-751).